Association of sociodemographic factors, lifestyle behaviors, anthropometric measures, and cardiometabolic health markers with blood pressure in adolescents: a cross-sectional analysis

Nascimento-da-Silva Fábio fabionascimento813@gmail.com 1
Valentim-Silva João Rafael 2
Meschiari César Arruda 3
Lima Ildercílio Mota de Souza 4
Guedes Dartagnan Pinto 5
Casonatto Juliano 1
1 Anhanguera Pitágoras Unopar University , Londrina , Paraná , Brazil
2 University of Vassouras , Saquarema , Rio de Janeiro , Brazil
3 Federal University of Acre , Rio Branco , Acre , Brazil
4 Federal University of Goiás , Goiânia , Goiás , Brazil
5 State University of Northern Paraná , Jacarezinho , Paraná , Brazil
Oliveira Sonia
Electronic publication date: 2025 Oct 13
Publication date: 2025
Volume: 13
Electronic Location ID: e20151
Received 2025 Apr 17; Accepted 2025 Sep 8
Copyright: ©2025 Nascimento-da-Silva et al.
Copyright year: 2025
Copyright holder: Nascimento-da-Silva et al.
License: This is an open access article distributed under the terms of the Creative Commons Attribution License, which permits unrestricted use, distribution, reproduction and adaptation in any medium and for any purpose provided that it is properly attributed. For attribution, the original author(s), title, publication source (PeerJ) and either DOI or URL of the article must be cited.
License URL: https://creativecommons.org/licenses/by/4.0/

Keywords: Adolescent health, Hypertension risk factors, Cardiometabolic outcomes, Anthropometric indicators, Socioeconomic disparities, Lifestyle behaviors analysis

Funding: No funding was received for this study.

==============================
Introduction

Evidence suggests that hypertension in adulthood may have its origins in childhood and adolescence. In this context, previous studies have demonstrated an association between lifestyle behaviors and nutritional status, both of which are linked to elevated blood pressure.

Objective

To examine the association between sociodemographic factors, subjective perception of health, lifestyle behaviors, anthropometric measurements, and cardiometabolic health markers with blood pressure in a sample of Brazilian adolescents.

Methods

A cross-sectional study was conducted with 273 adolescents (56% girls), aged 14–20 years, from Rio Branco, Acre, Amazon Region of Brazil. Data were collected using self-reported questionnaires and direct measurements, including anthropometric indices, cardiometabolic biomarkers, and blood pressure. Data were analyzed using chi-square tests and binary logistic regression models. Linear regression, adjusted by all variables, including dimension (sub-category) and age, was used to verify if the independent variables were associated with elevated blood pressure among adolescents.

Results

Significant associations were observed between elevated blood pressure and the male sex (OR = 2.56; 95% CI [1.11–5.86], p = 0.026) and rural residency (OR = 3.65; 95% CI [1.44–9.21], p = 0.006). Aerobic activity presented a significant association with elevated blood pressure (OR = 8.04; 95% CI [1.04–61.65], p = 0.045). Waist circumference increased the likelihood of elevated blood pressure (OR = 5.43; 95% CI [1.34–21.94], p = 0.018). Cardiometabolic markers, such as fasting glucose, demonstrated a significant association with elevated blood pressure (OR = 2.43; 95% CI [1.00–5.90], p = 0.048). No significant associations were found for muscle-strengthening activity, screen time, sleep duration, or food intake.

Conclusion

Our findings emphasize the crucial role of sociodemographic factors, especially the male sex and rural residency, in predicting elevated blood pressure among adolescents. Adolescents who perform the recommended amount of aerobic activity and who present acceptable fasting glucose values during adolescence are less likely to present high blood pressure. Waist circumference together with other anthropometric measurements might be a useful indicator for screening elevated blood pressure risk. These findings highlight the necessity for early detection, targeted interventions, and personalized strategies to reduce the hypertension risk and improve cardiometabolic health in the young population.

Introduction

Early exposure to risk factors through inadequate lifestyle choices, particularly during adolescence, directly contributes to the increasing prevalence of cardiometabolic diseases, including heart disease and diabetes mellitus (GBD, 2020). Adolescence is a crucial life stage, marked by growing autonomy and the formation of long-lasting lifestyle behaviors, that significantly influence health in adulthood (National Academies of Sciences , 2019).

Chronic non-communicable diseases, such as hypertension, are among the leading causes of mortality worldwide, including in Brazil. The prevalence of hypertension in Brazil has risen alarmingly, from 21.4% in 2013 to 23.9% in 2019, affecting approximately 38.1 million individuals (Brazil, 2019b). Evidence suggests that hypertension in adulthood may have its origins in childhood and adolescence, as children and adolescents with high blood pressure are more likely to develop hypertension later in life (Gartlehner et al., 2020).

Sedentary behavior, along with insufficient physical activity, excessive screen time, inadequate sleep, poor dietary habits, and unfavorable anthropometric measures, contributes to adverse cardiometabolic outcomes. Longitudinal studies show that 81% of adolescents do not meet the recommended levels of physical activity during the transition from adolescence to adulthood, which contributes to a reduction of about one-quarter in their physical activity (Corder et al., 2019). Additionally, adolescents tend to spend more time in sedentary activities, such as screen time (Van Sluijs et al., 2021), which is inversely related to sleep duration (Sehn et al., 2024).

Previous studies demonstrated the association between sedentary behavior, nutritional status, and obesity, all of which are linked to elevated blood pressure (Jeong & Kim, 2024). Healthy eating habits, such as consuming fruits and vegetables while avoiding high-fat foods and sugary drinks, can help to prevent serious target organ damage in children and adolescents, as well as obesity (Couch et al., 2021), which is associated with elevated levels of high sensitivity C-reactive protein, a marker of inflammation (Fujii et al., 2015). Additionally, obesity-related alterations in serum lipoproteins contribute to vascular alterations in childhood (Llop et al., 2023). As such, anthropometric measures, such as waist circumference and the waist to height ratio, are strong indicators of cardiovascular risk in adolescents (Pazin et al., 2020).

Considering the above, it would be interesting to expand the current scarce knowledge about the effects of selected sociodemographic factors, lifestyle behaviors, and cardiometabolic health markers on the blood pressure of adolescents who live in the Amazon region of Brazil. This understanding may aid in the formulation of more effective public health policies, as well as in providing information for the adequate allocation of resources for health promotion and the prevention of cardiometabolic harm in the present and future health of youths.

In this context, the objective of the current study was to examine the associations between sociodemographic factors, subjective perception of health, lifestyle behaviors, anthropometric measurements, and cardiometabolic markers with blood pressure in a sample of Brazilian adolescents. The hypothesis was immediately established in the sense that adolescents who report risk behaviors and who present altered cardiometabolic health markers tend to present more elevated blood pressure than their peers with healthier lifestyle behaviors.

Materials & Methods

Study design

This was an observational, cross-sectional study, conducted in accordance with the Strengthening the Reporting of Observational Studies in Epidemiology (STROBE) guidelines (Von Elm et al., 2008). The data analyzed were derived from two educational institutions located in Rio Branco, Acre, in the Amazon region of Brazil. Due to the longitudinal nature of the associated projects, which focused on the implementation of health education programs within the school environment, the study involved adolescents enrolled exclusively in these schools. The intervention protocols received approval from the Research Ethics Committee of the Federal University of Acre (Brazil Platform–CAAE number: 66876122.3.0000.5010).

Participants

The sample consisted of adolescents aged 14–20 years from two educational institutions in Rio Branco, Acre, Amazon region of Brazil. Recruitment occurred between November and December 2023. The students participated voluntarily in the experiment, after receiving authorization from their parents or guardians. To this end, all students enrolled at the beginning of the 2023 school year and their parents or guardians were contacted and informed about the study’s nature and objectives, the principle of confidentiality, and the non-influence on academic performance, and invited to participate in the data collection. Refusal to participate in the study or failure to respond to the invitation after three attempts made on different days and times were considered sample losses.

Exclusion criteria included temporary health issues (e.g., fractures, bronchitis) or permanent conditions (e.g., chronic diseases, physical disabilities) that might hinder participation; (b) use of any type of medication that could induce changes in study variables (cytostatic steroids); (c) not following prior fasting recommendations (10 to 12 h); (d) pregnancy or lactation; and (e) being on a specific diet. The rights of all participants were safeguarded by the Free and Informed Consent Form signed by the students and their parents or guardians.

Data collection

The project included a validated Brazilian version of the Youth Risk Behavior Survey (Guedes & Lopes, 2010), covering demographic data, health perception, and lifestyle behaviors. Additionally, anthropometric measurements, blood pressure, and cardiometabolic markers were performed. Data were collected between November and December 2023 by a team of researchers who were familiar with the instrument and trained in its procedures. The questionnaire was answered at a single moment, individually by each of the participants, and in the place and time of their class. The mean time taken to complete the questionnaire was 40 min. Assessors were not blinded to participant characteristics.

Regarding demographic data, in addition to sex and age, information was collected on the area where the young person lived, obtained through the Brazil Environmental Registry (Brazil, 2023), and on parents’ schooling and family economic status, according to the Brazil Classification Criteria (Brazil, 2019a). The subjective perception of health was assessed using the question: In general, how do you view your health? The response options were “excellent”, “very good”, “good”, “bad”, and “very bad”. Based on the responses provided by the adolescents, two groups were dichotomized for analysis: (a) low subjective perception of health, attributed to adolescents who reported “good,” “bad,” or “very bad”; and (b) high subjective perception of health, attributed to those who reported “excellent” or “very good.”.

Lifestyle behaviors were assessed using items related to physical activity, sedentary behavior, sleep duration, and dietary intake. Physical activity was evaluated through two specific questions: (a) “Over the past seven days, on how many days did you engage in physical activity for at least 60 minutes? (Include any type of activity that increased your heart rate and breathing, such as brisk walking, running, cycling, swimming, or similar activities; the total time counts, meaning it does not have to be 60 consecutive minutes—you may add up the time spent on physical activity throughout the day)”; and (b) “Over the past seven days, on how many days did you perform muscle-strengthening activities (such as push-ups, sit-ups, resistance band exercises, or weight training)?” Response options for both questions ranged from “none” to “seven days.”

According to international public health guidelines for physical activity (Bull et al., 2020), specifically in the aerobic component, participants were classified into two groups: recommended aerobic physical activity (7 days/week) and insufficient aerobic physical activity (<7 days/week). In the case of the muscle strengthening component, participants were also classified into two groups: recommended muscle-strengthening physical activity (≥ 2 days/week) and insufficient muscle-strengthening physical activity (<2 days/week).

Sedentary behavior was addressed by assessing recreational screen time, through the following questions: (a) “In a typical or usual week, how many hours do you watch TV?”; and (b) “In a typical or usual week, how many hours do you use a desktop computer, laptop, tablet, or smartphone for any activity that is unrelated to some type of work or schoolwork? “A predefined time scale was provided to answer both questions, in which respondents indicated their choice from among six categories, ranging from “<1 hour/day” to “>5 hours/day”. The questions separately considered the use of screen devices equivalent to weekdays and weekends (Saturday and Sunday). A weighted average of weekday and weekend data was used to identify daily recreational screen time. Based on this, two groups were dichotomized for analysis: (a) recommended sedentary behavior (≤ 5 hours/day of recreational screen time); and (b) high sedentary behavior (≥ 6 hours/day of recreational screen time).

Data on sleep duration were collected considering school days and weekends, using a typical or usual week as a reference, through four structured questions: “On school days: (a) what time do you usually go to sleep? (b) and what time do you wake up?”. “On weekend days (Saturday and Sunday): (a) what time do you usually go to sleep? (b) and what time do you wake up?” Based on the reports presented by the adolescents, sleep time on weekdays and weekends was calculated. A weighted average of data from weekdays and weekends was used to identify sleep duration per night. For the purpose of analysis, sleep duration was stratified into two groups: (a) recommended sleep duration (≥ 8 hours/night); and (b) insufficient sleep duration (<8 hours/night) (Hirshkowitz et al., 2015).

Regarding food intake, participants reported how frequently they consumed fruits/vegetables, fats and oils, soft drinks, processed meats, and sweetened products through the following questions: “In the last seven days, how often have you: (a) eaten fruits and/or vegetables (don’t consider fruit juices)?; (b) consumed fats and oils (fried food, French fries, or similar)?; (c) drunk soft drinks (coke, Pepsi, or similar)?; (d) consumed sausage meat, hot dog, or similar?”; and (e) consumed cakes, pies, cookies, sweets, or similar?”. The response options for these questions were: “none”; “1–3 times in the last seven days”; “4–6 times in the last seven days”; and “≥ 4 times/day”. Based on the responses, food intake was dichotomized into two groups for analysis: (a) low intake (“none,” and “1–3 times in the last seven days”; and (b) high intake (“4–6 times in the last seven days” and “≥ 4 times/day”) (Guedes & Lopes, 2010).

Anthropometric data consisted of height, body weight, and waist circumference measures, in line with the methodology proposed by the World Health Organization (De Onis & Habicht, 1996). The body mass index was calculated as the ratio between weight in kilograms and height in meters (kg/m2), and the waist/height ratio as the waist circumference divided by height, expressed in centimeters. Based on BMI, the anthropometric nutritional status of the students was classified into two categories, according to the cut-off points for sex and age suggested by the International Obesity Task Force–IOFT (Cole et al., 2000): eutrophic and overweight/obesity. Waist/height ratio values were used to analyze abdominal fat accumulation, adopting a cut-off point of 0.50 (Ashwell & Hsieh, 2005).

Systolic (SBP) and diastolic (DBP) blood pressure were measured following the guidelines of the American Academy of Pediatrics (Flynn et al., 2017), using the auscultatory method with a mercury sphygmomanometer. Measurements were taken on the left arm with the adolescent seated and after a minimum rest period of five minutes. Two readings were obtained, and the average of these was used for analysis. Blood pressure values were then dichotomized based on the updated classification of blood pressure categories and stages for children and adolescents, using percentiles or age-specific cut-off points for those aged 13 years and older (Flynn et al., 2017).

The serum metabolic markers of the adolescents metabolic markers were identified through plasmatic measurements of total cholesterol (TC), high sensitivity C-reactive protein (hs-CRP), and fasting glucose (GLU). Blood samples were analyzed at the reference laboratory, and quality control was based on the criteria from the Clinical Pathology Society. Plasmatic measurements were performed by collecting 10 mL of venous blood samples at the elbow crease after a 10–12-h fasting period, between 7:00 and 8:00 a.m. The serum was immediately separated by centrifugation; TC values were measured by the enzymatic colorimetric test (Roche, Indianapolis, IN, USA; Modular Analyzer); GLU levels by the hexokinase method using a Siemens ADVIA 2400, and hs-CRP concentrations by means of the quantitative immuno-nephelometry method (BN II, Siemens, Munich, Germany). Values of TC < 170 mg/dL and GLU < 100 mg/dL were considered acceptable according to the American Diabetes Association (American Diabetes Association, 2022), while values of CRP < 1.0 mg/dL were categorized as normal, based on the criteria of the American Heart Association and the Centers for Disease Control and Prevention (Pearson et al., 2003).

Statistical analysis

Sample size was estimated using G*Power 3.1 software, assuming a medium effect size (OR = 1.5), α = 0.05, and power = 80%, which indicated a minimum required sample of 246 participants. Our final analytical sample (n = 273) exceeded this requirement, ensuring adequate statistical power. Chi-square tests were employed to analyze associations between blood pressure (dependent variable) and independent variables, with statistically significant predictors further evaluated using logistic regression models. Independent variables with a significance level of p < 0.200 were analyzed using logistic regression to estimate the magnitude of the associations with the dependent variable, presented as odds ratios (OR) and their corresponding 95% confidence intervals (CI). The threshold of p < 0.200 was used during the bivariate analysis to reduce the risk of Type II error and avoid prematurely excluding variables that may be important confounders or predictors in the multivariate model. This approach allows for purposeful selection of variables, where initially inclusive criteria help identify all potentially relevant variables. One model was applied: a regression model, adjusted by all variables, including the dimension (sub-category) and age, accounting for the independent variables’ dimensions. Collinearity diagnostics were performed, and no evidence of significant correlation among the independent variables was detected, as all tolerance values were greater than 0.1 and all variance inflation factors (VIFs) were below 10. A significance threshold of p ≤ 0.050 was used to determine statistical significance. All analyses were conducted using the IBM® SPSS® Statistics for Windows Package, version 29 (IBM Corporate, Armonk, NY, USA).

Results

From a target population of 1,040 adolescents, we were able to invite 1,032 participants to take part in the study, as six individuals were excluded due to positive responses on the PAR-Q questionnaire and two indigenous students could not be evaluated owing to ethical restrictions. Among those invited, 131 adolescents declined to participate, and 439 did not respond to the invitation, a situation potentially influenced by the timing of data collection at the end of the school year. During initial screening, 40 adolescents were excluded due to health conditions and 16 due to continuous medication use. On the day of blood collection, 27 participants did not attend and were not assessed, and four pregnant participants were not evaluated. Additionally, 102 adolescents had incomplete data, as the assessments were conducted on different days and some participants were absent for certain measurements. Consequently, the final analytical sample comprised 273 adolescents (Fig. 1).

Figure 1 Flowchart of the study sampling process.

Table 1 summarizes the distribution of normal and elevated blood pressure across different sociodemographic factors, subjective perception of health, lifestyle behaviors, anthropometric measurements, and cardiometabolic health markers, along with the chi-square test results.

Table 1 Blood pressure prevalence rates (CI 95%) with stratification for sociodemographic factors, subjective perception of health, lifestyle behaviors, anthropometric measurements, and cardiometabolic health markers in a sample of Brazilian adolescents.

		Blood pressure	Chi-square test	
	n = 273	Normal
% (CI 95%)	Elevated
% (CI 95%)	χ 2	p-value	
Demographic data	
Sex						
Girls	153	91.7% (88.5–94.9)	8.3% (5.1–11.5)	6.123	0.013	
Boys	120	84.2% (79.9–88.5)	15.8% (11.5–20.1)	
Age						
14–16 years	173	88.4% (84.6–92.2)	11.6% (7.8–15.4)	0.438	0.508	
17–20 years	100	91.0% (87.6–94.4)	9.0% (5.6–12.4)	
Living area						
Urban	240	91.7% (88.5–94.9)	8.3% (5.1–11.5)	10.961	0.001	
Rural	33	72.7% (67.5–77.9)	27.3% (22.1–32.5)	
Parents’ schooling						
Low	77	92.2% (89.0–95.4)	7.8% (4.6–11.0)	0.905	0.341	
High	196	88.3% (84.5–92.1)	11.7% (7.9–15.5)	
Household economic class						
Low	165	92.7% (89.6–95.8)	7.3% (4.2–10.4)	4.930	0.026	
High	108	84.3% (80.0–88.6)	15.7% (11.4–20.0)	
Subjective perception of health	
Good/bad/very bad	173	87.3% (83.4–91.2)	12.7% (8.8–16.6)	2.181	0.140	
Very good/excellent	100	93.0% (90.0–96.0)	7.0% (4.0–10.0)	
Lifestyle behaviors	
Aerobic PA						
Recommended	47	97.9% (96.2–99.6)	2.1% (0.4–3.8)	4.315	0.038	
Insufficient	226	87.6% (83.7–91.5)	12.4% (8.5–16.3)	
Muscle-strengthening PA						
Recommended	105	89.5% (85.9–93.1)	10.5% (6.9–14.1)	0.004	0.950	
Insufficient	168	89.3% (85.7–92.9)	10.7% (7.1–14.3)	
Recreational screen time						
Recommended	70	94.3% (91.6–97.0)	5.7% (3.8–9.6)	2.389	0.122	
High	203	87.7% (83.8–91.6)	12.3% (8.4–16.2)	
Sleep duration						
Recommended	175	87.4% (83.5–91.3)	12.6% (8.7–16.5)	1.950	0.163	
Insufficient	98	92.9% (89.3–95.5)	7.1% (4.1–10.1)	
Fruit/vegetable intake						
Low intake	154	89.0% (83.3–92.7)	11.0% (7.3–14.7)	0.064	0.800	
High intake	119	89.9% (86.4–93.4)	10.1% (6.6–13.6)	
Soft drink intake						
Low intake	185	90.3% (86.8–93.8)	9.7% (6.2–13.2)	0.482	0.488	
High intake	88	87.5% (83.6–91.4)	12.5% (8.6–16.4)	
Processed meat intake						
Low intake	236	89.0% (85.3–92.7)	11.0% (7.3–14.7)	0.285	0.593	
High intake	37	91.9% (88.7–95.1)	8.1% (4.9–11.3)	
Anthropometric measurements	
Body mass index						
Eutrophic	210	94.8% (92.2–97.4)	5.2% (2.6–7.8)	27.790	<0.001	
Overweight/obesity	63	71.4% (66.1–76.7)	28.6% (23.3–33.9)	
Waist/height ratio						
≤ 0.50	245	91.8% (88.6–95.0)	8.2% (5.0–11.4)	15.219	<0.001	
>0.50	28	67.9% (62.4–73.4)	32.1% (26.6–37.6)	
Waist circumference						
Acceptable	208	95.7% (93.3–98.1)	4.3% (1.9–6.7)	36.471	<0.001	
High	65	69.2% (63.8–74.6)	30.8% (25.4–36.2)	
Cardiometabolic markers	
Total cholesterol						
Acceptable	182	87.9% (84.1–91.7)	12.1% (8.3–15.9)	1.235	0.267	
High	91	92.3% (89.2–95.4)	7.7% (4.6–10.8)	
HDL						
Acceptable	243	89.7% (86.1–93.3)	10.3% (6.7–13.9)	0.261	0.610	
Low	30	86.7% (82.7–90.7)	13.3% (9.3–17.3)	
LDL				0.098	0.755	
Acceptable	249	89.6% (86.0–93.2)	10.4% (6.8–14.0)	
High	24	87.5% (83.6–91.4)	12.5% (8.6–16.4)	
C-reactive protein						
Acceptable	216	91.2% (87.9–94.5)	8.8% (5.5–12.1)	3.635	0.057	
High	57	82.5% (78.0–97.0)	17.5% (13.0–22.0)	
Fasting glucose						
Acceptable	222	91.0% (87.6–94.4)	9.0% (5.6–12.4)	4.400	0.036	
High	46	80.4% (75.7–85.1)	19.6% (14.9–24.3)	
Notes.

Bold values indicate variables with p < 0.200, the cutoff used for inclusion in the logistic regression analysis to estimate the magnitude of associations. PA: physical activity; HDL: high density lipoprotein; LDL: low density lipoprotein.

Bivariate analyses showed a significant association between blood pressure and sex (p = 0.013), with boys being more likely to present elevated blood pressure compared to girls. Additionally, a significant association was observed between blood pressure and living area (p = 0.001); individuals living in rural areas were more likely to present elevated blood pressure compared to those living in urban areas. Household economic class was significantly associated with blood pressure (p = 0.026), with a higher prevalence of elevated blood pressure among participants from higher economic classes compared to those from lower economic classes. Conversely, no significant association was found between blood pressure and age or parents’ schooling.

Subjective perception of health also did not demonstrate a significant association with blood pressure. Aerobic activity was significantly associated with blood pressure (p = 0.038). No significant associations were observed between blood pressure and muscle-strengthening activity, sedentary behaviors measured as recreational screen time, sleep duration, and fruit/vegetable, soft drink, or processed meat intake.

Regarding anthropometric measurements, a significant association was observed between blood pressure and the body mass index (p < 0.001), with individuals classified as overweight or obese more likely to present elevated blood pressure compared to those with normal weight. Similarly, waist circumference and the waist to height ratio also showed a significant association with blood pressure (p < 0.001), indicating that adolescents with a higher concentration of centripetal fat were more likely to present elevated blood pressure than those with normal values. Concerning cardiometabolic markers, a significant association was observed between blood pressure and fasting glucose (p = 0.036). No significant associations were observed between blood pressure and total cholesterol, high-density lipoprotein, low-density lipoprotein, and high sensitivity C-reactive protein levels.

Figure 2 illustrates the associations between blood pressure, sex, family economic class, living area, sleep duration, recreational screen time, aerobic activity, waist/height ratio, waist circumference, overweight/obesity, glucose, and high sensitivity C-reactive protein.

Figure 2 Adjusted regression model between blood pressure, sex, household economic class, living area, sleep duration, recreational screen time, aerobic activity, waist/height ratio, waist circumference, overweight/obesity, glucose, and high sensitivity C-reactive.

The association remained significant for the male sex (OR = 2.56; 95% CI [1.11–5.86], p = 0.026) and rural living area (OR = 3.65; 95% CI [1.44–9.21], p = 0.006). No longer statistically significant after adjustment for family economic class (OR = 1.95; 95% CI [0.86–4.40], p = 0.106). These findings suggest that the male sex and rural living area independently contribute to the likelihood of elevated blood pressure.

The associations for insufficient sleep (OR = 0.438; 95% CI [0.17–1.10], p = 0.079) and recreational screen time (OR = 2.16; 95% CI [0.70–6.66], p = 0.176) were no longer statistically significant after adjustment. However, the association between insufficient aerobic activity and elevated blood pressure remained significant (OR = 8.04; 95% CI [1.04–61.65], p = 0.045). No statistical significance was observed after adjustment for subjective perception of health (OR = 0.51; 95% CI [0.21–1.25], p = 0.145).

The associations for the waist/height ratio (OR = 1.11; 95% CI [0.36–3.45], p = 0.848) and overweight/obesity (OR = 1.80; 95% CI [0.44–7.32], p = 0.411) were no longer statistically significant after adjustment. However, the association between waist circumference and elevated blood pressure remained significant (OR = 5.43; 95% CI [1.34–21.94], p = 0.018).

The likelihood of elevated blood pressure remained non-significant for high sensitivity C-reactive protein (OR = 2.32; 95% CI [0.98–5.46], p = 0.053), while the association between fasting glucose risk and elevated blood pressure remained significant (OR = 2.43; 95% CI [1.00–5.90], p = 0.048).

Discussion

The current study examined the association between sociodemographic factors, subjective perception of health, lifestyle behaviors, anthropometric measurements, and cardiometabolic markers with blood pressure in a sample of Brazilian adolescents.

The results indicated that adolescents of the male sex, who live in a rural area, who perform insufficient aerobic physical activity, and present altered waist circumference and fasting glucose values were associated with elevated blood pressure.

The findings showed that sex, especially being male, is associated with high blood pressure. This corroborates other studies, which reported that male adolescents (13–17 years), similarly to the population of the current study, had a higher prevalence of high blood pressure (11.3%) compared to female adolescents (5.7%) (Hardy et al., 2021). Previous data demonstrate that many factors could contribute to this scenario, starting in childhood, including body composition, sex steroids, and heart rate regulation (Nagata et al., 2022).

Other findings in the current work indicated that adolescents living in rural areas were more likely to exhibit elevated blood pressure. While previous studies have explored some of these relationships (Chen et al., 2023), the present study uniquely analyzed the combined influence of adolescent lifestyle behaviors on blood pressure, as opposed to studying these factors in isolation. Notably, the magnitude of association analysis revealed no significant relationships between blood pressure and the variables age, parental education, household economic class, subjective perception of health, muscle-strengthening activity, recreational screen time, sleep duration, food intake, body mass index, waist/height ratio, total cholesterol, high-density lipoprotein, low-density lipoprotein, and high sensitivity C-reactive protein levels.

Living in a rural area was significantly associated with elevated blood pressure, with 26.5% of rural participants affected compared to 9.1% of urban participants. These findings align with prior research linking rural residency to a higher prevalence of hypertension, often attributed to limited healthcare access, dietary habits, and socioeconomic disparities. Similarly, other studies have reported higher systolic blood pressure values in boys across all ages, with girls from rural areas showing a higher prevalence of elevated blood pressure (13.3%) compared to their urban counterparts (7.2%) (Krzywinska-Wiewiorowska et al., 2017).

Our findings further revealed that, after adjusting for other variables, participants living in rural areas had three to four times higher odds of elevated blood pressure (OR = 3.65; 95% CI [1.44–9.21]). These results reinforce the role of sociodemographic factors, particularly rural residence, in influencing blood pressure. Previous studies also highlighted this association, with one study reporting a value of 26.1% of rural children presenting elevated blood pressure (Krzywinska-Wiewiorowska et al., 2017), consistent with the prevalence observed in the current study. Together, these findings underscore the need for targeted interventions in rural areas to address the underlying factors contributing to elevated blood pressure. No significant association was found between parents’ schooling, household economic class, and blood pressure in our sample.

In line with data available in the literature (Silva et al., 2018), the results of the current study showed that adolescents who reported <7 days/week of aerobic physical activity presented higher elevated blood pressure. These findings may be partially explained by the frequency of physical activity; adolescents who were more physically active were less likely to present high blood pressure. Most studies have shown that aerobic physical activity is a protective factor against high blood pressure. In this context, school-based physical education programs represent a key structured opportunity to promote regular engagement in vigorous activities. Recent findings indicate that adolescents who actively participate in these programs exhibit significantly better cardiometabolic health and a lower prevalence of elevated blood pressure (Benavente-Marín et al., 2024a). Other research reported a positive relationship between physical activity and lower blood pressure in boys (Chen et al., 2023). No significant associations were found among participants with a low subjective perception of health, 12.7% had elevated blood pressure, compared to 7.0% among those with a high subjective perception of health.

Recent evidence underscores that vigorous-intensity physical activity may exert stronger protective effects on blood pressure levels than lower-intensity modalities. Adolescents who engage frequently in vigorous activity demonstrate lower rates of hypertension and better overall cardiometabolic profiles (Benavente-Marín et al., 2024b).

Regarding screen time, prior studies reported a linear increase in blood pressure prevalence among boys with a higher TV viewing time, rising from 8.5% with less than 2 h to 15.8% with over 4 h daily (Oliveira et al., 2018). However, in the current study, screen time was not significantly associated with blood pressure. A recent systematic review and meta-analysis confirmed that recreational screen time is associated with elevated blood pressure in children and adolescents, and that recreational screen time significantly increased the odds ratio. However, after subgrouping the results according to age, the positive association between recreational screen time and elevated blood pressure only remained in children, not adolescents, suggesting that children at lower ages are more at risk of elevated blood pressure than adolescents (Farhangi et al., 2023).

Similarly, although previous studies have linked shorter sleep duration to higher blood pressure in adolescents (Wang et al., 2023), our findings did not support this association. Despite this, studies have suggested an explanation linked to circadian misalignment and that increased sympathetic nervous system activity could change cardiometabolic factors (Confortin et al., 2022).

Additionally, screen time may be associated with other health behaviors, such as reduced adherence to healthy diets, particularly among adolescents with less parental supervision or whose parents have lower educational attainment. Studies have shown that high screen time, coupled with low parental education, can negatively influence eating patterns and increase cardiometabolic risk (Wärnberg et al., 2021).

In terms of food intake, our findings align with earlier research that found no significant relationship between dietary patterns and blood pressure (Chen et al., 2023). A possible explanation for this result is that adolescents generally remain metabolically healthy through mid-adolescence, including adolescents with obesity (Phillips, 2017) and the impact of food intake on specific cardiometabolic indicators, such as blood pressure, may require a longer period to manifest (Winpenny, Sluijs & Forouhi, 2021).

The absence of significant associations between food intake and blood pressure may be explained by several factors, including the limitations of self-reported dietary data, the short recall period employed (seven days), and the possibility of delayed effects of dietary habits on specific cardiometabolic markers such as blood pressure. Furthermore, many adolescents—despite exhibiting poor dietary behaviors—may still present a metabolically healthy profile during mid-adolescence, particularly if they have not yet experienced the effects of long-term cumulative exposure (Phillips, 2017; Winpenny, Sluijs & Forouhi, 2021).

Regarding anthropometric measurements, initially scores equivalent to the waist/height ratio were shown to be closely associated with blood pressure. The waist/height ratio is a sensitive predictor of hypertension in adolescents (Cheah et al., 2018). However, studies have reported other measurements as being better for predicting elevated blood pressure, compared to the waist/height ratio. For example, in a cross-sectional study with children and adolescents aged 7–17 years, overweight/obesity was a better predictor of elevated blood pressure in both sexes (Dong et al., 2015).

Another anthropometric measurement, waist circumference, also showed an association with high blood pressure. In the current study, adolescents with a high waist circumference were five to six times more likely to have high blood pressure, thus, a high waist circumference might indicate an increased risk of elevated blood pressure, and the health risks of these adolescents would be underestimated if screening by one anthropometric measurement alone. These findings suggest that measurement of waist circumference could represent an assistive tool when used together with overweight/obesity and the waist/height ratio to identify high blood pressure risks (Zhang, Zhao & Chu, 2016).

Initially, our results showed that overweight and/or obese adolescents were significantly more likely to present high blood pressure, and previous studies found that overweight/obesity was a good predictor of elevated blood pressure in adolescents (Taghizadeh et al., 2021). However, overweight/obesity exhibited low sensitivity in discriminating adolescents with elevated blood pressure (Abbaszadeh et al., 2017). This could be due to the inability of the body mass index to measure fat distribution and differentiate adipose tissues and muscle mass (Tee, Gan & Lim, 2020). Thus, screening by overweight/obesity alone could potentially lead to underestimation of obesity-related diseases, including hypertension, in adolescents.

After adjustment of the regression model between blood pressure and dependent variables (high sensitivity C-reactive protein and fasting glucose), fasting glucose was significantly associated with higher blood pressure.

In the current study, adolescents with elevated fasting glucose values were two to three times more likely to present high blood pressure. Studies with adolescents aged 10 to 19 years reported a 46.4% prevalence of high blood pressure in those with elevated fasting glucose levels, which increases the cardiometabolic risks considering an association with overweight or obesity (Vasudevan et al., 2022).

Levels of the other cardiometabolic variable evaluated, high sensitivity C-reactive protein, were not associated with obesity and higher waist circumference, although findings have linked high sensitivity C-reactive protein and blood pressure to cardiovascular disease risk (Casagrande & Lawrence, 2024). These findings may be associated with the sample size and the lack of a specific cut-off point for adolescents.

The current study presents some limitations that should be acknowledged. Although quality control procedures were implemented, self-reported lifestyle data may be subject to recall and social desirability bias. Potential selection bias is acknowledged, given the high non-response rate at recruitment. Some participants had incomplete data, which may have influenced results. Sensitivity analyses were not performed, which may limit robustness of findings. Although the initial target population was 1,040 adolescents, only 273 were analyzed. This reduction may limit statistical power and affect the generalizability of findings, especially for borderline associations. Finally, the cross-sectional design precludes causal inference. Despite these limitations, the strengths of the study include consideration of diverse sociodemographic and anthropometric variables, which enhance the reliability and generalizability of the findings.

Conclusions

In conclusion, the results of the current study showed a positive association between socioeconomic factors and high blood pressure, especially in male adolescents, who live in rural areas. Furthermore, adolescents who perform the recommended amount of aerobic activity and who present acceptable fasting glucose levels during adolescence are less likely to present high blood pressure. Our findings suggest waist circumference, used together with other anthropometric measurements, might be a useful indicator for screening elevated blood pressure risk in the school setting or in routine primary-level health services for adolescents who live in the Amazon region of Brazil. Thus, educational and public health interventions are needed, aimed at promoting the early detection of risk factors for elevated blood pressure and healthy lifestyle behaviors among adolescents. The importance of the study findings is highlighted due to the fundamental role of targeted interventions and personalized strategies to reduce hypertension risk and improve cardiometabolic health in the young population.

From a public health perspective, our findings underscore the need to implement regular blood pressure screening programs in schools, particularly in rural areas. Intervention strategies should encourage consistent physical activity and incorporate simple anthropometric measures—such as waist circumference—as cost-effective tools for the early identification of cardiometabolic risk among adolescents. These measures could facilitate routine monitoring in both school-based and primary health care settings.

Supplemental Information

Supplemental Information 1 Raw data with all study variables

All variables used for statistical analysis to examine the associations between all variables with blood pressure.

Supplemental Information 2 English-language codebook

All codes used in raw data to identify all variables.

We extend our heartfelt gratitude to the adolescents of the Dom Pedro II Military High School—Rio Branco Unity and the Federal Institute of Education, Science and Technology of Acre–Rio Branco Campus, who voluntarily participated in this study, contributing to the data collection of 273 subjects from Rio Branco, Acre. Their enthusiastic and generous involvement made this research possible. We also appreciate the support of the local communities and institutions that facilitated our efforts, the Secretary of State for Education, Federal University of Acre, and Clinical Analysis Laboratory–Women’s and Children’s Health Care System of Acre. Their contributions were invaluable to the advancement of knowledge in this field.

Additional Information and Declarations

Competing Interests

Author Contributions

Human Ethics

Data Availability

The authors declare there are no competing interests.

Fábio Nascimento-da-Silva conceived and designed the experiments, performed the experiments, analyzed the data, prepared figures and/or tables, authored or reviewed drafts of the article, and approved the final draft.

João Rafael Valentim-Silva analyzed the data, authored or reviewed drafts of the article, and approved the final draft.

César Arruda Meschiari performed the experiments, authored or reviewed drafts of the article, and approved the final draft.

Ildercílio Mota de Souza Lima performed the experiments, authored or reviewed drafts of the article, and approved the final draft.

Dartagnan Pinto Guedes conceived and designed the experiments, analyzed the data, prepared figures and/or tables, authored or reviewed drafts of the article, and approved the final draft.

Juliano Casonatto conceived and designed the experiments, analyzed the data, prepared figures and/or tables, authored or reviewed drafts of the article, and approved the final draft.

The following information was supplied relating to ethical approvals (i.e., approving body and any reference numbers):

Federal University of Acre (66876122.3.0000.5010) approved this study.

The following information was supplied regarding data availability:

The raw measurements are available in the Supplementary File.

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
