# Peer review of "Association of sociodemographic factors, lifestyle behaviors, anthropometric measures, and cardiometabolic health markers with blood pressure in adolescents: a cross-sectional analysis"

_PeerJ, doi:10.7717/peerj.20151_

## Round 0.1 · original submission · Major Revisions

Reviewer 1 ·

Basic reporting

The paper examined the association between SES, behavioural, nutrition and blood pressure in a sample of adolescent. Several comments to be considered.
1. Over 700 adolescents were excluded. Whether there are differences in basic characteristics for the included and the excluded? It would be great if a supplementary table could be provided.
2. For linear regression, whether the collinearity has been examined?
3. For P values, I assume two decimals are enough.

Experimental design

The research questions were well-defined.

Validity of the findings

The external validity needs to be examined in another study.

·

Basic reporting

The manuscript is generally written in clear, professional English, but minor grammatical issues (e.g., repeated words such as "a a significative") should be corrected. The introduction is well developed and provides sufficient context for the public health significance of adolescent hypertension. Relevant literature is appropriately cited, and the structure of the manuscript follows PeerJ standards.

Figures and tables are relevant, clearly labeled, and informative. The inclusion of a flowchart (Figure 1) and stratified data (Table 1) facilitates understanding. It is noted that raw data are available and referenced, fulfilling data transparency requirements.

Experimental design

The study uses a cross-sectional observational design appropriate for the purpose of the study. A sample of 273 adolescents is adequately described and exclusion criteria are clearly stated. Ethical approval has been verified and is in compliance with PeerJ policies.

Data collection methods for lifestyle behaviors, anthropometry, and biochemical parameters are well documented and follow accepted standards. The use of STROBE guidelines is commendable and increases methodological transparency.

Validity of the findings

Statistical analyses are robust and appropriate for the dataset. Bivariate and multivariate models were used effectively. Logistic regression models were implemented correctly, and confidence intervals support the robustness of the reported associations. The results are interpreted with caution within the limitations of a cross-sectional design, and causal claims are avoided. The findings that male gender, rural residence, and waist circumference are risk factors for hypertension are reasonable and consistent with the existing literature.

Additional comments

This study contributes significantly to public health research, particularly in underrepresented populations, in adolescence. Examining a wide range of lifestyle and biochemical markers provides a valuable holistic picture of BP determinants. The article would be improved if the following recommendations were addressed:

- Grammar and language fluency should be improved throughout the article.

- Discussion of null findings (e.g., no association with screen time or food intake) should be expanded.

- Conclusions should discuss public health implications and potential screening/intervention strategies more explicitly.

Reviewer 3 ·

Basic reporting

The manuscript is well written. However, there are occasional typographical and grammatical errors. Some phrasing could also be made more concise and fluid. I recommend some polishing for clarity.

The literature cited is appropriate. Key studies are mentioned, though a few claims (e.g., relationships with screen time) could benefit from more up-to-date references. I suggest you add this reference: DOI: 10.3390/jcm10040795 to highlight how screen time and parental education can influence health behaviors (like adherence to healthy diet), potentially affecting cardiometabolic risk, including blood pressure. This could for example be included in the discussion section lines 333-350).
Moreover, to strengthen the argument about the importance of adhering to physical activity guidelines, specifically regarding vigorous intensity I recommend this reference: DOI: 10.7717/peerj.16815. This could be included in the discussion section lines 325-330).

There is evidence showing that engaging in school-based physical education is crucial for maintaining cardiometabolic health in children, highlighting the importance of structured opportunities for vigorous activity in preventing hypertension (DOI: 10.7717/peerj.16990). This could also be included in the discussion section (lines 325–330).

Tables and figures are relevant. However, the figure 2 should improve in the design, increase the font letters in the figures.

Experimental design

The study addresses an important public health issue—adolescent hypertension—using a sample from an underrepresented region (Amazon, Brazil). It adds valuable data, though its cross-sectional nature limits causal interpretation.

Data collection instruments (questionnaires, measurements) and variable classifications are adequately described. However, regarding the sample size: Although the final sample size (n=273) is reported, the initial sample and reasons for exclusions (losses of 759) raise concerns about selection bias. This issue should be more explicitly addressed.

Regarding the statistical Analysis: The use of chi-square tests and logistic regression is appropriate. The choice of p<0.20 as the threshold for including variables in multivariate models should be justified.

Validity of the findings

The findings are generally supported by the data, but the wide confidence intervals for some associations indicate potential issues with sample size or variability. Also potential biases from self-reported data and sample size should be discussed more thoroughly in the limitation section.

While not highly novel, the study contributes useful regional data and integrates multiple factors (sociodemographic, behavioral, biological).

---

## Round 0.2 · Minor Revisions

Dear authors,

Thank you for your interesting submission to PeerJ. Before we can move to production, please make sure to proofread your manuscript and also review and answer to the last comments submitted by the reviewers.

Together, I ask you confirm and to make sure that:

- study design: clearly specify (even in text) whether the study was prospective, retrospective, or cross-sectional (I think I saw it only in the title), and clearly define the observational design. Make sure that the flow diagram has no mistakes.
- participant recruitment and eligibility: make sure that inclusion / exclusion criteria are explicitly stated in the methods, not just tables or figures; clarity source population (hospital / clinic / community) and recruitment period; indicate how many participants were excluded and reasons.

- sample size justification: did G*power but detail assumptions used (expected effect size, power, alpha level) & make clear whether the final sample met or exceeded this requirement.

- data collection and variables: make sure to include details on how exposures, outcomes, and covariates were measured or abstracted; clarify if data collection was standardized and whether assessors were blinded; cite or describe if or how questionnaires used were validated.

- statistics: report how normality and variance assumptions were tested prior or applying parametric vs non-parametric tests; for regression models, specify whether multicollinearity, confounding and model fit were assessed; state any adjustments for multiple comparisons, if relevant.

- bias and limitations: make sure to address comprehensively potential sources of bias (selection, recall, missing data handling, etc); clarify whether sensitivity analyses were performed to test robustness of findings.

Many thanks in advance.

**Language Note:** The review process has identified that the English language must be improved. PeerJ can provide language editing services - please contact us at [email protected] for pricing (be sure to provide your manuscript number and title). Alternatively, you should make your own arrangements to improve the language quality and provide details in your response letter. – PeerJ Staff

Reviewer 1 ·

Basic reporting

The paper is well written, and the English is acceptable. The structure is well organised, and the results are clear to me.

Experimental design

It is within the aims and scope of the journal. The research question is well defined and meaningful. No technical issue was detected, and a knowledge gap was identified.

Validity of the findings

The findings were convincing. No issue specified.

Additional comments

The paper was revised, and I have no further comment.

·

Basic reporting

The article is generally written in clear, academic, and professional English. However, some sentences (e.g., “Aerobic activity presented a significative association…”) still contain grammatical errors. “Significant” should be used instead of “Significative.”

The introduction is adequately supported by the literature, and the existing knowledge gap is well identified.

Figures, tables, and references are presented accurately. The tables are clear and well-structured.

All data and analysis methods are clearly explained; however, better labeling of the Supplementary Raw Data file is recommended.

Experimental design

The research question is clear and meaningful: The relationship between blood pressure and lifestyle and sociodemographic variables in adolescents.

An appropriate cross-sectional design was used, following the STROBE guidelines.

Participant selection, inclusion/exclusion criteria, and ethical approvals were clearly stated.

However, although the sample size initially appeared large (1040 participants), it was reduced to only 273 participants in the final analysis. The effects of this sample loss should be further discussed in terms of statistical power.

The measurements used (blood pressure, waist circumference, glucose, etc.) were obtained using valid and reliable methods.

Validity of the findings

The findings are presented logically and relate well to the original research question.

Regression analyses were conducted appropriately, and interactions and confounding factors were considered.

However, some results were statistically borderline (e.g., p=0.053 for CRP), so authors should approach these findings with caution in their interpretations.

The clinical relevance of predictors such as waist circumference and fasting glucose is emphasized.

Additional comments

The authors have largely responded to previous comments and made significant revisions to the article.

The introduction has been expanded, making the literature more comprehensive.

The results section is strong in its public health implications.

Recommendation: Professional English editing is recommended for linguistic corrections.

Additionally, the description of the comparison between the waist/height ratio and BMI based on the level of statistical significance could be further elaborated.

It would be beneficial to label the data with clearer explanations in the supplementary materials section.

---

## Round 0.3 · accepted · Accept

Dear authors,

I think that all issues pointed out have been addressed now. Thank you for your work and submission. I am accepting your manuscript for publication in PeerJ. All the best,
Sonia

[#